# Amentoflavone Promotes Cellular Uptake and Degradation of Amyloid-Beta in Neuronal Cells

**DOI:** 10.3390/ijms23115885

**Published:** 2022-05-24

**Authors:** Byung Hee Han, Brooke Cofell, Emily Everhart, Courtney Humpal, Sam-Sik Kang, Sang Kook Lee, Jeong Sook Kim-Han

**Affiliations:** 1Department of Pharmacology, Kirksville College of Osteopathic Medicine, A.T. Still University of Health Sciences, Kirksville, MO 63501, USA; sa204482@atsu.edu (B.C.); eeverhart@atsu.edu (E.E.); sa203744@atsu.edu (C.H.); jeongkimhan@atsu.edu (J.S.K.-H.); 2College of Pharmacy, Seoul National University, Seoul 08826, Korea; sskang@snu.ac.kr (S.-S.K.); sklee61@snu.ac.kr (S.K.L.)

**Keywords:** Alzheimer disease, amyloid-beta clearance, cellular uptake, scavenger receptors, neurodegeneration, flavonoids, polyphenols, drug discovery

## Abstract

Deposition of fibrillar forms of amyloid β-protein (Aβ) is commonly found in patients with Alzheimer’s disease (AD) associated with cognitive decline. Impaired clearance of Aβ species is thought to be a major cause of late-onset sporadic AD. Aβ secreted into the extracellular milieu can be cleared from the brain through multiple pathways, including cellular uptake in neuronal and non-neuronal cells. Recent studies have showed that the naturally-occurring polyphenol amentoflavone (AMF) exerts anti-amyloidogenic effects. However, its effects on metabolism and cellular clearance of Aβ remain to be tested. In the present study, we demonstrated that AMF significantly increased the cellular uptake of both Aβ_1-40_ and Aβ_1-42_, but not inverted Aβ_42-1_ in mouse neuronal N2a cells. Though AMF promoted internalization of cytotoxic Aβ_1-42_, it significantly reduced cell death in our assay condition. Our data further revealed that the internalized Aβ is translocated to lysosomes and undergoes enzymatic degradation. The saturable kinetic of Aβ uptake and our pharmacologic experiments showed the involvement of receptor-mediated endocytosis, in part, through the class A scavenger receptors as a possible mechanism of action of AMF. Taken together, our findings indicate that AMF can lower the levels of extracellular Aβ by increasing their cellular uptake and clearance, suggesting the therapeutic potential of AMF for the treatment of AD.

## 1. Introduction

Alzheimer’s disease (AD) is a progressive neurodegenerative disease, causing dementia and cognitive impairment in the elderly [1,2,3,4,5]. Neuropathological features of AD include accumulation of fibrillar amyloid β-protein (Aβ) in the brain parenchyma as neuritic plaques, and in the brain vessels as cerebral amyloid angiopathy [6,7,8]. Aβ peptides (commonly Aβ_1-40_ and Aβ_1-42_) are produced via enzymatic cleavage of the amyloid precursor protein and secreted into the brain extracellular milieu [9]. The levels of Aβ in the brain are tightly controlled by maintaining balance between its biosynthesis and clearance in healthy individuals. For example, approximately 8% of the total Aβ is newly biosynthesized per hour and cleared at an equal rate from the human brain [10]. Overproduction of Aβ is responsible for the development of the early-onset familial forms of AD associated with deposition of fibrillar forms of Aβ species, neuroinflammation, neurodegeneration, and cognitive deficit [1,11]. In contrast, impaired Aβ clearance appears to be a major cause of Aβ accumulation in late-onset sporadic AD, which accounts for over 95% of cases [11].

The major mechanisms underlying clearance of extracellular Aβ species in the brain include Aβ removal to the peripheral blood and the lymphatic system, enzymatic degradation by extracellular proteinases, and clearance by cellular uptake within the CNS tissues [12]. There are multiple types of cells involved in cellular uptake and clearance of both soluble and fibrillar forms of Aβ species through receptor-mediated endocytosis and other endocytic pathways. For example, soluble forms of the Aβ species can be taken up by neurons via the low-density lipoprotein receptor-related protein (LRP-1)-dependent endocytosis in astrocytes [13], vascular smooth muscle cells [14], and neurons [15]. Heparan sulfate proteoglycans (HSPGs) are a family of extracellular matrix proteins that are known to mediate cellular Aβ uptake in neuronal cells [16]. In addition, cell surface scavenger receptors such as macrophage scavenger receptor 1 (SCAR1), macrophage receptor with collagenase structure (MARCO), and CD36 are involved in binding and cellular uptake of soluble and toxic forms of Aβ species in microglia and immune cells (reviewed in [12,17]).

Amentoflavone (AMF) is a naturally-occurring biflavonoid (an apigenin dimer) present in plants such as *Selaginella tamariscina* [18]. There are growing lines of evidence that AMF affords anti-amyloid activities via multiple functions. First, we and others found that AMF exhibits distinct pharmacologic actions among its similar biflavonoids and monoflavonoids on inhibiting the fibrillization of Aβ and disaggregating preformed Aβ fibrils [19,20]. Our in vitro study revealed that AMF has a high binding affinity for Aβ fibrils and destabilizes the β-sheet structure of Aβ fibrils, leading to the formation of amorphic Aβ aggregates [19]. Molecular docking simulation indicates that AMF and its similar bioflavonoids are docked to the N-terminal amino acid residues of Aβ fibrils and disrupt intramolecular hydrogen bonding [21,22]. Second, AMF exhibits a potent antioxidant property and can chelate Cu^2+^ (another risk factor of AD), which reduces reactive oxygen species (ROS) formation and neurotoxicity induced by the Aβ-Cu^2+^ complex [22]. Third, recent in vivo studies show that AMF protects the brain against Aβ -induced neurotoxicity and cognitive impairment through the modulation of the AMPK/GSK3β signaling pathway [23,24]. Taken together, these findings indicate that AMF exerts the anti-amyloidogenic effect and neuroprotection through multiple mechanisms. However, the effects of AMF on Aβ metabolism and cellular clearance remains to be examined.

In the present study, we demonstrate that AMF has the most potent effect among the test compounds on promoting the cellular uptake of soluble Aβ in N2a neuronal cells and glial cells. In addition, we report that AMF increases the cellular uptake of Aβ via receptor-mediated endocytosis and subsequently the lysosomal degradation without causing cytotoxicity. These results suggest that AMF provides a novel class of anti-amyloid agents that can prevent amyloid deposition in the CNS by facilitating the cellular uptake and clearance of Aβ.

## 2. Results

### 2.1. Amentoflavone Most Potently Increases Cellular Uptake of Aβ_1-40_ in N2a Neurons

To examine the effects of biflavonoids on cellular uptake of Aβ_1-40_ in neuronal N2a cells, we chose structurally similar amentoflavone-type biflavonoids: amentoflavone (AMF), ginkgetin (GGT), isoginkgetin (IGG), and sciadopitysin (SDP) (Figure 1A). Cellular uptake of Aβ_1-40_ was assessed after 3 h incubation with 200 nM Aβ_1-40_ conjugated with a fluorophore 5-carboxytetramethylrhodamine (TAMRA-Aβ_1-40_) in the presence of various concentrations of the test compounds. We found all the biflavonoids increased the intracellular accumulation of the fluorescently labeled TAMRA-Aβ_1-40_ in a concentration-dependent manner (Figure 1B). Moreover, AMF revealed the most potent activity in promoting the cellular internalization of TAMRA-Aβ_1-40_ in N2a cells. There was an increase in Aβ_1-40_ uptake by 1.48 ± 0.05, 2.48 ± 0.18, or 3.64 ± 1.12-fold when N2a cells were treated for 3 h with 1, 3, or 10 μM amentoflavone, respectively (Figure 1C). We found that apigenin and quercetin (which contain a single flavonoid unit), and the green tea polyphenol, (-)-epigallocatechin gallate (EGCG) had little to no effect on promoting the cellular Aβ_1-40_ uptake (Figure 1C). Next, to cytologically confirm that AMF increased cellular uptake of Aβ_1-40_, N2a cells were incubated with 200 nM TAMRA-Aβ_1-40_ in the presence or absence of 10 μM AMF for 3 h and subjected to confocal microscopy. We observed that the internalized TAMRA-Aβ_1-40_ was present in the cytosol of N2a cells in the absence of AMF (Figure 1D, the middle row). Treatment of N2a cells with AMF markedly increased the intracellular TAMRA-Aβ_1-40_ signal in the cytosol of N2a cells (Figure 1D, the bottom row).

### 2.2. Amentoflavone Increases Cellular Uptake of Both Aβ_1-40_ and Aβ_1-42_ without Causing Cytotoxicity

To rule out the possibility that the chemical modification of Aβ_1-40_ by the addition of the fluorophore TAMRA moiety to the peptide affects the internalization process of Aβ_1-40_, we performed the cellular uptake assay using Aβ_1-40_ labeled with another fluorophore, 5-hydroxyfluorescein (FAM-Aβ_1-40_) (Figure 2). Consistently, we found that AMF increased the intracellular accumulation of FAM-Aβ_1-40_ in a concentration-dependent fashion in N2a cells (Figure 2A). Co-treatment of N2a cells with FAM-Aβ_1-40_ and the monoclonal anti-Aβ_1-40_ antibody HJ2 significantly lowered the cellular uptake of FAM-Aβ_1-40_ in the presence or absence of AMF (*p* < 0.05) vs. the control IgG-treated group (Figure 2B). Similar to the Aβ_1-40_, the Aβ_1-42_ peptide was also taken up by N2a cells when FAM-Aβ_1-42_ was used in the assay (Figure 2C).

Moreover, co-treatment of cells with 10 μM AMF significantly increased cellular uptake of FAM-Aβ_1-42_ compared with the group without AMF (7.52 ± 0.50 vs. 3.34 ± 0.84 pmole/mg protein; *p* < 0.05, Figure 2C). However, the cellular uptake of the reverse Aβ peptide, FAM-Aβ_42-1_ was almost undetectable in the presence and absence of amentoflavone when the uptake assay was performed for 3 h (*p* > 0.05; Figure 3C). To determine whether the increased cellular uptake of Aβ peptide consequently caused cytotoxicity, N2a cells were incubated with Aβ in the presence of various concentrations of AMF for 24 h, and cytotoxicity was assessed by the levels of lactate dehydrogenase (LDH) released into the culture media. We found FAM-Aβ_1-40_ at a concentration of 200 nM did not cause cell death in the absence or presence of AMF (*p* > 0.05, Figure 2D). However, treatment of the cells with 200 nM FAM-Aβ_1-42_ for 24 h significantly increased the LDH release (O.D.: 0.27 ± 0.03) vs. the control group (O.D.: 0.27 ± 0.03; *p* < 0.05). This FAM-Aβ_1-42_-induced LDH release was significantly inhibited by co-treatment of N2a cells with AMF at all the concentrations tested (1, 3, and 10 μM) vs. the group treated with FAM-Aβ_1-42_ alone (*p* < 0.05; Figure 2E). Since Aβ_1-42_ showed the cytotoxicity in our experimental condition and is prone to aggregate rapidly into the fibrillar forms [19], we used Aβ_1-40_ species in the rest of the present study.

### 2.3. Internalized Aβ_1-40_ Peptide Undergoes Lysosomal Degradation in N2a Cells

We sought to examine the possible mechanisms by which AMF promotes the cellular uptake of Aβ_1-40_. We first determined the time-course of Aβ internalization and binding to the extracellular matrix (Figure 3) by incubating N2a cells with 200 nM TAMRA-Aβ_1-40_ in the presence or absence of 10 μM AMF for up to 6 h. The accumulation of the intracellular TAMRA-Aβ_1-40_ load appeared to reach a plateau by 6 h. Co-treatment of N2a cells with AMF significantly increased the internalization of Aβ_1-40_, which appeared to be saturated after 6 h (Figure 3A). To measure the levels of the cell membrane-bound form of Aβ_1-40_, the TAMRA fluorescent intensity liberated from the extracellular matrix by trypsin digestion was measured and plotted (Figure 3B). We found that TAMRA-Aβ_1-40_ binding to the extracellular side of the plasma membrane reached a plateau within 3 h after the incubation. AMF significantly increased the TAMRA-Aβ_1-40_ binding to cell membrane after one, three and six hours of incubation, which was saturated by three hours after the incubation. Conversely, the concentrations of TAMRA-Aβ_1-40_ in the culture medium significantly lowered in the presence (170 ± 1.6 nM) vs. in the absence of 10 μM AMF (189 ± 0.6 nM; *p* < 0.05; Figure 3C) after 3 h of the incubation.

To examine the integrity of Aβ_1-40_ taken up into neuronal cells, N2a cells were treated with the lysosomal enzyme inhibitor, leupeptin (100 μg/mL), unlabeled Aβ_1-40_ peptide in the presence or absence of AMF and for 3 h, and cell lysates were subjected to SDS-PAGE followed by immunoblotting with an anti-Aβ_1-40_ antibody (Figure 4A,B). We observed that the intracellular levels of the full-length (~4-kDa), monomeric Aβ_1-40_ was increased in the presence of AMF (Figure 4A,B). Co-treatment of the cells with leupeptin significantly increased the accumulation of the internalized Aβ_1-40,_ indicating the involvement of proteolytic degradation of intracellular Aβ_1-40_ by lysosomal enzymes. The internalized Aβ_1-40_ was almost completely degraded within 3 h when N2a cells were incubated with Aβ_1-40_ for 3 h followed by incubation in the Aβ_1-40_-free culture medium (washout) for an additional 3 h (Figure 4A,B). Leupeptin did not influence AMF-induced promotion of Aβ_1-40_ uptake (Figure 4C).

To further investigate the subcellular localization of Aβ_1-40_ taken up by neurons in the presence of AMF, N2a cells were subjected to immunofluorescent double-labeling with the lysosomal marker LAMP-1 and confocal microscopy (Figure 5). In the absence of 10 μM AMF, red fluorescent intensity from TAMRA-Aβ_1-40_ was colocalized with the lysosomal marker LAMP-1 (green) in the perinuclear region of N2a neurons. Treatment of N2a cells clearly increased the punctate deposition of the TAMRA-Aβ_1-40_ which colocalized to the lysosomal LAMP-1 signal.

### 2.4. Evidence That AMF Promotes Aβ_1-40_ Uptake via the Receptor-Mediated Endocytosis

To examine which receptor-mediated endocytic pathway is responsible for the AMF-induced promotion of Aβ_1-40_ uptake, the TAMRA-Aβ_1-40_ uptake assay was performed in the presence pharmacologic inhibitors of each receptor-mediated endocytic pathway. We found that fucoidan, the ligand for scavenger receptor class A (both MARCO and SRA1) [25,26,27,28], had no effect on TAMRA-Aβ_1-40_ uptake without AMF; however, 100 μM fucoidan significantly decreased the TAMRA-Aβ_1-40_ uptake in the presence of AMF (1.98 ± 0.17 vs. 3.01 ± 0.22 pmole/mg protein, *p* < 0.05; Figure 6A). Next, to examine the role of heparan sulfate glycoproteins (HSPGs) in AMF-mediated Aβ_1-40_ uptake, the uptake assay was performed in the presence or absence with the HSPG inhibitor, heparin [16,29]. We found 20 U/mL heparin in part attenuated TAMRA-Aβ_1-40_ uptake in the presence or absence of AMF (Figure 6B). Next, we performed the Aβ_1-40_ uptake assay utilizing ApoE particles purified from astrocytes, which is known to inhibit the LRP-1-mediated endocytosis of Aβ in astrocytes [13]. We found that the treatment of N2a cells with 200 nM ApoE particles markedly decreased the cellular uptake of TAMRA-Aβ_1-40_ by ~75%, whereas this ApoE particle inhibited AMF-induced TAMRA-Aβ_1-40_ uptake by ~30% (Figure 6C). Lastly, we observed that the Aβ_1-40_ uptake with or without AMF treatment was not blocked by cytochalasin D that is known to inhibit pinocytic endocytosis of Aβ [25].

### 2.5. AMF Promotes Aβ_1-40_ Uptake in Microglia and Astrocytes

We performed the Aβ uptake assay using the TAMRA-Aβ_1-40_ peptide in murine BV2 microglia and astrocytes (Figure 7). We found these types of cells were capable of taking up Aβ_1-40_. In BV2 microglia (Figure 7A), treatment with 2 μM and 10 μM AMF significantly increased the intracellular levels of TAMRA-Aβ_1-40_ by 2.2-fold and 3.0-fold, respectively. Likewise, the cellular uptake of TAMRA-Aβ_1-40_ in astrocytes was also significantly increased to a similar extent (Figure 7B).

## 3. Discussion

The results in the present study reveal multiple lines of evidence that the biflavonoid AMF promotes the internalization of extracellular Aβ which subsequently undergoes enzymatic degradation in neuronal cells. Utilizing the fluorescently labeled Aβ peptides, we demonstrated that AMF amongst its similar biflavonoids and other polyphenolic compounds possessed the most profound effect on the cellular uptake of both Aβ_1-40_ and Aβ_1-42_, but not inverted Aβ_42-1_. Though AMF increased cellular transport of cytotoxic Aβ_1-42_, it significantly attenuated Aβ-induced neuronal cell death in our assay condition. Our data further revealed that the internalized Aβ is translocated to lysosomes where this peptide undergoes enzymatic degradation. The saturable kinetic of Aβ uptake suggests the involvement of receptor-mediated endocytosis. Moreover, we found that the class A scavenger receptors appear to contribute to Aβ uptake facilitated by AMF. Taken together, our findings indicate that AMF can lower the levels of extracellular Aβ species by increasing their cellular uptake and clearance, suggesting the therapeutic potential of AMF for the treatment of AD.

In the present study, we first explored the structure-activity relationship of amentoflavone-type biflavonoids on the cellular uptake of Aβ utilizing the fluorescently labeled Aβ_1-40_ [14,30]. This TAMRA-Aβ_1-40_ serves as a powerful tool to quantify the levels of Aβ uptake within cells and to monitor subcellular localization of the internalized peptide (Figure 1). We found that AMF, which has all three hydroxyl groups in the R1, R2, and R3 positions (Figure 1A), had most potently increased the cellular uptake of Aβ. However, the substitution of these hydroxyl groups with one or more methoxy groups lowered their activity in promoting Aβ uptake. These results are in line with our previous report that AMF among its similar biflavonoids possesses the most potent effect on inhibiting the aggregation of Aβ and promoting disaggregation of Aβ fibrils [19]. Our computational analysis study indicates that AMF and other similar biflavonoids directly bind to the N-terminal amino acid residues of Aβ fibrils via the π-π interactions [21,22]. Moreover, we found that AMF promotes disaggregation of Aβ fibrils by forming hydrogen bonds between the hydroxyl groups at the R2 and R3 and Aβ peptides, leading to the disruption of the β-sheets of the fibril [21]. We found that monoflavonoids such as apigenin and quercetin had no effect on the cellular uptake of Aβ in our assay condition (Figure 1). The green tea polyphenol, (−)-epigallocatechin gallate (EGCG), is known to have anti-amyloidogenic effects through its direct interaction with the Aβ pathways and the neurodegenerative cascades (reviewed in [31]). However, this polyphenol had little to no effect on Aβ uptake (Figure 1C). Interestingly, both quercetin and EGCG are considered as “pan assay interference compounds (PAINS)” that can interfere with cell-free and cell-based assays via multiple mechanisms due to their nature of chemical structure [32,33,34]. They contain a PAINS motif, catechol, which is responsible for the false positive results in certain biochemical assay systems, in particular utilizing fluorescent substrates. However, both quercetin and EGCG did not cause false positive signals in our Aβ uptake assay (Figure 1C). The PAINS database search [33] reveals that AMF and the other biflavonoids we used in the present study do not contain any potential PAINS motifs. Taken together, our findings indicate the genuine action of AMF on the intracellular transport of Aβ. It may be possible that a direct physical interaction of AMF with Aβ peptides changes the 3-D structure of Aβ, which is more favorable to cellular uptake than that of a free form of Aβ. Since we and others found that AMF possesses a variety of biological activities [35,36,37,38], it is also possible that AMF’s actions are due to changes in the cellular transport machinery involved in the Aβ clearance pathway. Further study is necessary to identify the precise mechanism underlying AMF’s action on the facilitation of Aβ uptake.

To rule out the possibility that AMF-induced facilitation of Aβ uptake was due to the chemical modification of Aβ by the addition of the TAMRA moiety, we used the second fluorophore FAM-labeled Aβ_1-40_ in the uptake assay. Consistently, AMF increased the cellular uptake of FAM-Aβ_1-40_, indicating that the chemical modification was not attributed to the internalization of Aβ (Figure 2A). The internalization of FAM-Aβ_1-40_ was markedly blocked by >70% in the presence of the anti-Aβ antibody HJ2 that binds to the C-terminal amino acid residues 34–40 of Aβ. These results suggest that the Aβ uptake is indeed a specific process that is led by the recognition of certain residues within both Aβ_1-40_ and Aβ_1-42_. This notion is further supported by our finding that there was an undetectable level of cellular Aβ uptake when the inverted Aβ peptide, FAM-Aβ_42-1,_ was used in our uptake assay. Likewise, AMF had no effect on the internalization of the inverted peptide, supporting the involvement of a selective transport mechanism of Aβ uptake in neuronal cells. Since Aβ species, especially Aβ_1-42,_ as forms of soluble monomers and oligomers, are neurotoxic, increased cellular uptake of Aβ_1-42_ might lead to N2a cell death. To test this possibility, we monitored an extracellular release of LDH, the marker for cytotoxicity after 24 h of incubation of neuronal cells with Aβ. We found that the cytotoxic Aβ_1-42_ peptide induced neuronal cell death in our assay condition, which was significantly attenuated in the presence of AMF (Figure 2E). Production of reactive oxygen species (ROS) plays a key role in mediating Aβ_1-42_-induced neuronal cell death [39], and AMF has a potent antioxidative action [22,35]. We therefore speculate that though there is an increased intracellular Aβ, AMF can protect the cells from Aβ-induced cytotoxicity via, at least, its antioxidative action.

We sought to further examine the molecular mechanisms by which AMF promoted cellular uptake of Aβ in neuronal cells. The internalization of different forms of Aβ species include a variety of selective and non-selective uptake processes in neurons and other types of cells [11,12,13,25]. Our time-course study revealed that both Aβ binding to the outer plasma membrane and its cellular uptake was saturable within 3–6 h, indicating the characteristic of a selective, receptor-mediated transport [30,40] in N2a neuronal cells. We have demonstrated that the internalized Aβ undergoes enzymatic degradation in lysosomes (Figure 4 and Figure 5). Accumulation of the full-length, monomeric Aβ (4 kDa) taken up by neuronal cells was greatly increased in the presence of the lysosomal enzyme inhibitor, leupeptin, whereas this internalized Aβ can be degraded within 3 h after the removal of leupeptin (Figure 4). Our complementary experiment further supports the lysosomal location of the internalized Aβ when assessed by immunolabeling followed by confocal microscopy (Figure 5). It is worth noting that the TAMRA moiety liberated from the Aβ peptide by enzymatic cleavage may not be exclusively retained within the lysosomal organelle. Nevertheless, the internalized TAMRA signals are mostly colocalized with the lysosomal marker, LAMP-1, in N2a cells (Figure 5). A previous study shows that soluble Aβ_1-40_ and Aβ_1-42_ can be internalized and targeted to different subcellular organelles [30]. Thus, further study is necessary to determine whether AMF has differential effects on the uptake and degradation of Aβ_1-40_ vs. Aβ_1-42_, or their soluble forms vs. oligomers vs. fibrils.

Various endocytic pathways (e.g., receptor-mediated endocytosis, phagocytosis, and pinocytosis), or non-endocytic pathways (energy-independent mechanism) contribute to the cellular uptake of Aβ depending on the cell contexts and Aβ species [14,25]. Regarding the receptor-mediated endocytosis, there are two major pathways: low-density lipoprotein receptor-related protein (LRP-1) [13,14,41] and heparan sulfate proteoglycans (HSPGs) in neurons and vascular smooth muscle cells [42,43,44,45,46]. Our results indicate both LRP1- and HSPG-dependent endocytic pathways play a major role in the cellular uptake of Aβ_1-40_ while there is a minimal role of the class A scavenger receptor-driven endocytosis in N2a cells (Figure 6). However, AMF appears to promote the cellular uptake of Aβ_1-40_ via the class A scavenger receptors as its blocker, fucoidan, could attenuate AMF-induced Aβ_1-40_ uptake in a concentration-dependent fashion (Figure 6). Since a family of the class A scavenger receptors include multiple members, including SCAR1 and MARCO, and fucoidan is a non-selective ligand for the receptors, further study is warranted to identify which receptor(s) is (are) responsible for AMF-induced Aβ uptake. This endocytic pathway seems to be ubiquitously present in other types of cells such as microglia and astrocytes (Figure 7), indicating its important role in the clearance of certain forms of Aβ species.

In conclusion, deposition of Aβ is a major risk factor that leads to neuroinflammation, neuronal cell loss, and cognitive decline in patients with AD. Impairment in the Aβ clearance pathway is responsible for accumulation of misfolded Aβ in the common late-onset sporadic AD. Secreted extracellular Aβ in the CNS can be cleared through multiple mechanisms, including cellular uptake by endocytosis and non-endocytic mechanisms. In the present study, we report that amentoflavone promotes cellular uptake of Aβ probably via the class A scavenger receptors. The internalized Aβ, in turn, undergoes lysosomal degradation without causing cytotoxicity in neuronal cells. Our results demonstrate that amentoflavone is a novel bioflavonoid that possesses such effects, suggesting the translational potential of amentoflavone and their derivatives to treat or slow the progression of brain impairment in patients with Alzheimer’s disease. Further study is necessary to determine if AMF’s action in promoting Aβ clearance also occurs in other experimental settings, including the mouse models of AD in vivo. Recently, there are growing lines of evidence showing that a variety of polyphenolic natural products exert the anti-amyloidogenic actions and improve cognitive performance via multiple mechanisms, including their anti-inflammatory and anti-oxidative activities (reviewed in [47,48,49,50,51]). It is therefore intriguing to further explore the effects of those polyphenols on the cellular uptake and clearance pathway of Aβ.

## 4. Materials and Methods

### 4.1. Materials

Mouse neuronal N2a (Neuro-2a) cells were purchased from American Type Culture Collections (ATCC, Rockville, MD, USA). Amentoflavone, ginkgetin, isoginkgetin, and sciadopitysin were prepared as described and their purity was greater than 98%, as determined by HPLC [19]. The monoclonal anti-Aβ antibody HJ2 and astrocyte-secreted apolipoprotein E (ApoE) particles [13] were kindly gifted by David Holtzman (Washington University in St. Louis, MO, USA). Leupeptin, heparin, fucoidan, and cytochalasin D were purchased from Sigma-Aldrich (St. Louis, MO, USA). Unlabeled and labeled human Aβ_1-40_ with 5-carboxytetramethylrhodamine (TAMRA-Aβ_1-40_), and TAMRA-Aβ_1-42_, and 5-carboxyfluorescein-labeled Aβ_1-40_ (FAM-Aβ_1-40_), FAM-Aβ_1-42_, and FAM-Aβ_42-1_ were obtained from AnaSpec (Fremont, CA, USA).

### 4.2. Cell Culture

N2a cells were grown in Dulbecco’s Modified Eagle Medium (DMEM) supplemented with 10% fetal bovine serum, 100 U/mL penicillin and 100 µg/mL streptomycin at 37 °C in a humidified CO_2_ incubator (Thermo Fisher Scientific, Waltham, MA, USA). Cells at less than five passages were used in the experiments. The immortalized BV-2 murine microglial cell line was grown and maintained in DMEM containing antibiotics and 2% fetal bovine serum [35]. The immortalized mouse astrocyte cell line was kindly gifted by David Holtzman at Washington University in St. Louis and maintained in DMEM containing antibiotics and 10% fetal bovine serum [52].

### 4.3. Aβ Uptake Assay

Fluorescent conjugated Aβ peptides were ≥95% in purity as determined by the HPLC method per the vendor (AnaSpec; Fremont, CA, USA). These peptides were dissolved in dimethyl sulfoxide to make 100 μM stock solutions, aliquoted, and stored at −80 °C until use. The fluorescent intensity and protein concentration of each peptide was determined as described below to calculate their specific activities (fluorescent intensity per mg of the conjugated peptide).

The Aβ uptake assay was performed as described, with modifications [13,25]. N2a cells were seeded in a 24-well culture plate (2 × 10^5^ cells/well) the day prior to the experiments. The next day (cell density at near confluency), cells were rinsed twice with warmed (37 °C) DMEM and incubated in DMEM containing 0.1% bovine serum albumin, 200 nM fluorescently labeled Aβ in the presence or absence of various concentrations of test compounds in a final volume of 200 µL. Cells were then incubated at 37 °C for three hours in the humidified CO_2_ incubator. The culture plate was placed on ice and the reaction medium was collected in a 1.5-mL microcentrifuge tube. Cells were rinsed twice with phosphate-buffered saline (PBS) containing 137 mM NaCl, 2.7 mM KCl, 10 mM Na_2_HPO_4_, and 1.8 mM KH_2_PO_4_ (pH: 7.4). To cleave cell surface-bound Aβ and extracellular matrix proteins, cells were treated with 0.05% trypsin in 200 µL PBS at room temperature on an orbital shaker for 5 min. Detached cells were transferred to a microcentrifuge tube and centrifuged at 5000× *g* for 5 min using a bench-top centrifuge (Thermo Fisher Scientific, Waltham, MA, USA). The supernatant and cell pellet were separated and kept at −80 °C until analysis. Cells were lysed in a buffer containing 20 mM HEPES (pH 7.4), 7.5 mM MgCl_2_, 1% sodium dodecyl sulfate, 1% Triton X-100, and Complete protease inhibitor cocktail (Sigma-Aldrich, St. Louis, MO, USA). The cell lysate was centrifuged at 10,000× *g* at 4 °C for 10 min. The fluorescent intensity was measured at λ_EX_ = 490 and λ_EM_ = 520 for FAM, and λ_EX_ = 545 and λ_EM_ = 575 for TAMRA using a Biotek Synergy HT plate reader (Winooski, VT). Protein concentrations of the cell lysates were determined using a BCA assay kit (Sigma-Aldrich, St. Louis, MO, USA). The fluorescent intensity of each sample was used to calculate an amount of Aβ in pmole and normalized to protein content. Cellular uptake of Aβ was expressed as pmole/mg protein.

To examine the role of receptor-mediated endocytosis on AMF-induced promotion of Aβ_1-40_ uptake, we performed the Aβ uptake assay utilizing pharmacologic inhibitors of endocytosis. N2a cells were pretreated with the ligand for scavenger receptor class A (both MARCO and SRA1), fucoidan (20 or 100 μg/mL) [25,26,27,28], the HSPG inhibitor, heparin (20 U/mL) [16,29], the inhibitor of LRP-1-mediated endocytosis ApoE particles (40 or 200 nM) [13], or the inhibitor of pinocytic endocytosis, cytochalasin D (0.4 or 2 μg/mL) [25]. Thirty minutes later, cells were subjected to the Aβ uptake assay with or without 10 μM AMF as described above.

### 4.4. Lactate Dehydrogenase (LDH) Activity Assay

The cytotoxicity of Aβ species in N2a cells was assessed using an LDH activity assay kit according to the manufacturer’s protocol (Sigma-Aldrich, St. Louis, MO, USA). Briefly, the Aβ uptake assay was performed as described above using FAM-Aβ_1-40_ or FAM-Aβ_1-42_. Twenty-four hours later, the incubation medium was transferred to a microcentrifuge tube and spun at 10,000× *g* for 5 min to remove cell debris. Triplicate samples of the assay medium were added with an equal volume of LDH master reaction mixture (50 µL/well) to each well of a 96-well plate. The optical density was measured at 37 °C for 30 min at a 5-min interval at a wavelength of 450 nm using a plate reader (Biotek, Winooski, VT, USA).

### 4.5. SDS-PAGE Electrophoresis and Immunoblotting

An equal amount of protein samples (5 µg/lane) was separated on a 4–20% gradient tris-tricine gel (Bio-Rad, Hercules, CA, USA) per our published protocol [35,53]. The gel was transferred to a nitrocellulose membrane (Bio-Rad) at 4 °C for 3 h. Blots were blocked with PBS containing 5% dry milk for 1 h and incubated in PBS solution containing an anti-Aβ antibody HJ2 (1:2000 dilution) or anti-α-tubulin antibody (1:5000 dilution, Sigma-Aldrich). After washing, blots were incubated with goat anti-mouse IgG conjugated with horseradish peroxidase (Bio-Rad, Hercules, CA, USA). The signal was visualized using the HRP chemiluminescent kit (ThermoFisher Scientific, Waltham, MA, USA), and photographic images were taken using a Syngene gel imaging system (Frederick, MD, USA).

### 4.6. Immunofluorescent Labeling and Confocal Microscopy

Poly L-lysine-coated coverslips (12 mm in diameter, Electron Microscopy Sciences, Hatfield, PA, USA) were placed in a 24-well plate and sterilized under the UV lamp. N2a cells were seeded on the coverslips (2 × 10^5^ cells/well) and grown in the cell culture incubator. The next day, the Aβ uptake assay was performed as described above using TAMRA-Aβ_1-40_. After three hours of incubation, cells were rinsed three times with 500 µL ice-cold PBS, and fixed with 4% paraformaldehyde for 20 min, followed by dehydration with 80% ethanol. Immunofluorescent labeling was performed per our published method [53,54]. Cells were incubated with a blocking solution containing 0.05% Triton X-100, 1% bovine serum albumin, and 0.2% dry milk in PBS for one hour. Cells were incubated overnight with an anti-LAMP-1 antibody (1:100 dilution, Sigma-Aldrich) at 4 °C and with goat anti-rabbit IgG conjugated with Alexa Flour 488 (1:1000 dilution, Thermo Fisher Scientific, Waltham, MA, USA). The coverslips were mounted on a microscope slide and fluorescent images were captured using a Leica TCS SP5 confocal microscope (Wetzlar, Germany).

### 4.7. Statistical Analysis

Data were expressed as mean ± SEM. Data were analyzed using one-way or two-way ANOVA followed by Tukey’s multiple comparison test using Prism GraphPad software version 8 (GraphPad, San Diego, CA, USA). A value of *p* < 0.05 was considered significant.

## Figures and Tables

**Figure 1 ijms-23-05885-f001:**
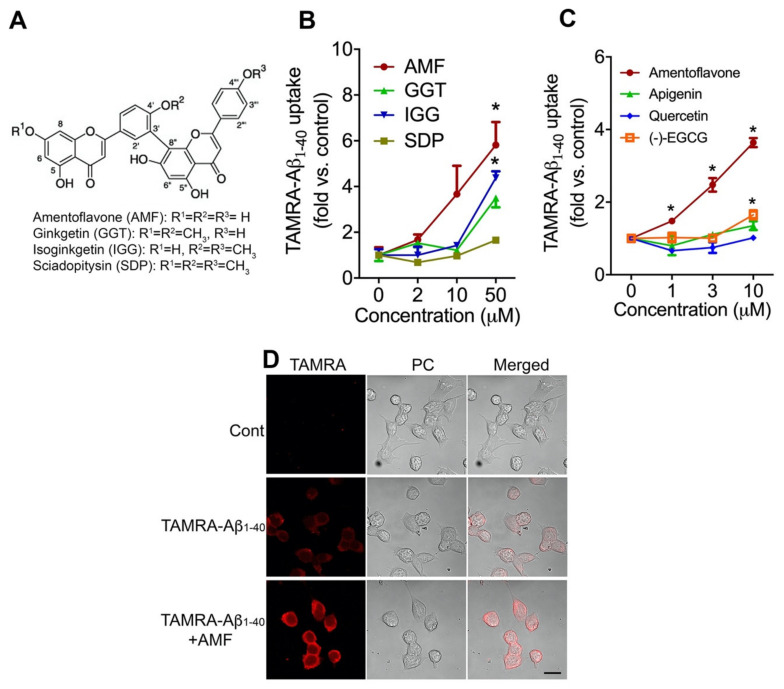
Amentoflavone shows the most potent effect on Aβ_40_ uptake in N2a neurons. (**A**). Chemical structure of amentoflavone-type biflavonoids used in the study. (**B**,**C**). Neuronal N2a cells were incubated with TAMRA-Aβ_1-40_ (200 nM) without or with various concentrations of flavonoids for 3 h at 37 °C in a CO_2_ incubator. Cells were harvested by trypsin treatment and the fluorescent intensity of the intracellular TAMRA signal was determined by fluorometry. Data represent mean ± S.E.M. (*n* = 4). *: *p* < 0.05 vs. control (no AMF treatment) as determined by two-way ANOVA followed by Tukey’s multiple comparison test. (**B**). AMF: amentoflavone, GGT: ginkgetin, IGG: isoginkgetin, and SDP: sciadopitysin. (**D**)**.** N2a cells were treated with TAMRA-Aβ_1-40_ (200 nM) with or without 10 µM amentoflavone (AMF) for 3 h at 37 °C. Cells were fixed with 4% paraformaldehyde and subjected to photographic imaging using a confocal microscope. Representative images of TAMRA and phase contrast (PC), and their merged images are presented. Scale bar: 10 µm.

**Figure 2 ijms-23-05885-f002:**
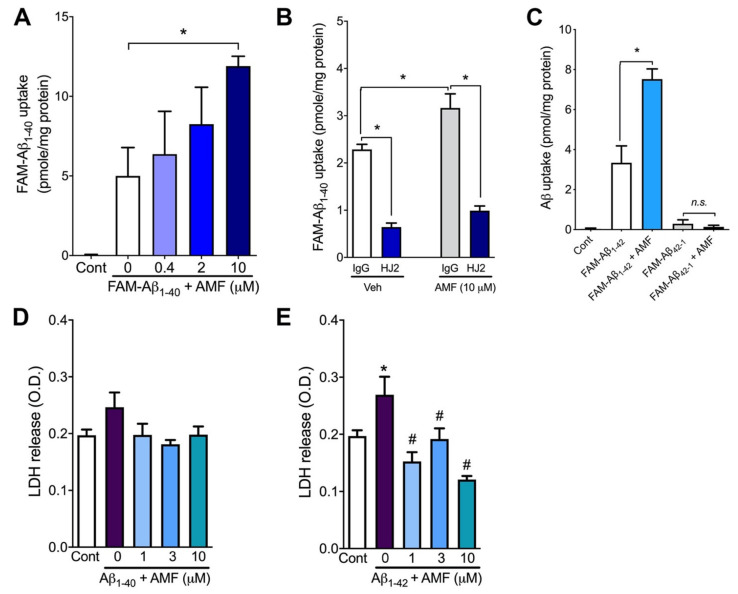
Amentoflavone promotes cellular uptake of both Aβ_1-40_ and Aβ_1-42_ without causing neurotoxicity. (**A**). N2a cells were incubated with 200 nM FAM-labeled Aβ_1-40_ (FAM-Aβ_1-40_) in the presence of various concentrations of amentoflavone (AMF). Three hours later, the intracellular FAM-Aβ_1-40_ load was determined. (**B**). The cellular Aβ_1-40_ uptake assay was performed by incubating 200 nM FAM-Aβ_1-40_ without (Veh) or with 10 µM AMF in the presence of control IgG or the anti-Aβ_1-40_ antibody, HJ2. (**C**). N2a cells were treated with 200 nM FAM-Aβ_1_-_42_ or the reverse Aβ peptide (FAM-Aβ_42-1_) in the presence or absence of 10 µM AMF for 3 h, and intracellular FAM load was determined. (**D**,**E**). N2a cells were incubated with or without 200 nM FAM-Aβ_1-40_ (**D**) or FAM-Aβ_1-42_ in the presence or absence of various concentrations of AMF. Twenty-four hours later, cytotoxicity was determined by the lactate dehydrogenase (LDH) assay in the culture medium. Data represent mean ± S.E.M. (*n* = 4). *: *p* < 0.05 as determined by one-way (**A**,**C**–**E**) or two-way (**B**) ANOVA followed by Tukey’s multiple comparison test (**A**–**C**). *: *p* < 0.05 vs. control; #: *p* < 0.05 vs. FAM-Aβ_1-42_ without AMF treatment (**E**).

**Figure 3 ijms-23-05885-f003:**
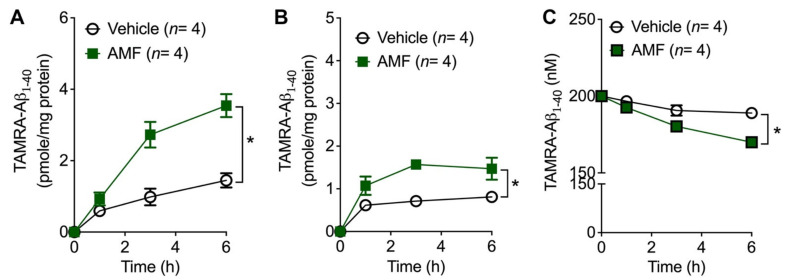
Time-course of cellular uptake of Aβ_1-40_. N2a cells were incubated in culture medium containing 200 nM TAMRA-Aβ_1-40_ alone (vehicle) or with 10 µM amentoflavone (AMF) for one, three and six hours. The TAMRA fluorescent intensity was measured in samples of cell lysate (**A**), bound to extracellular plasma membrane (**B**), and culture medium (**C**), Data indicate mean ± S.E.M. (*n* = 4). *: *p*< 0.05 vs. vehicle group assessed by two-way repeated measures ANOVA.

**Figure 4 ijms-23-05885-f004:**
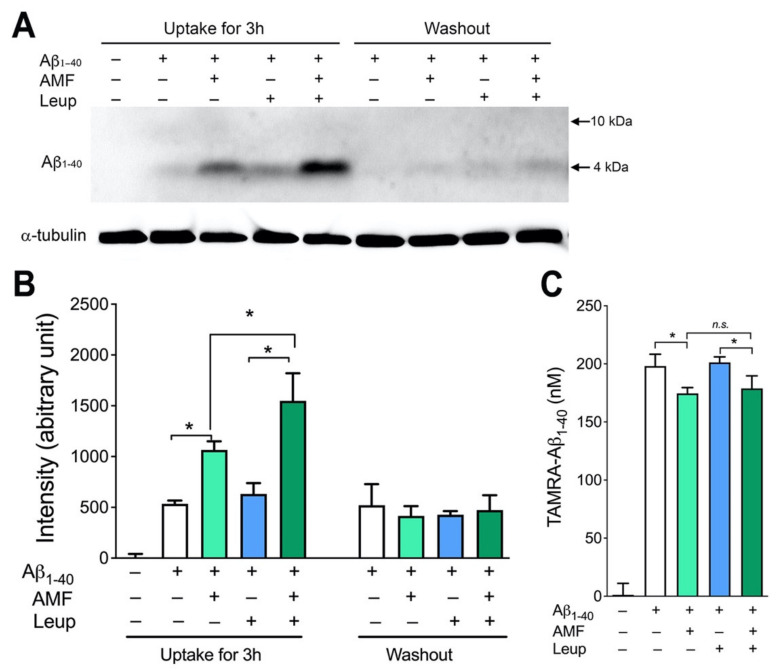
Amentoflavone promotes lysosomal degradation of internalized Aβ_1-40_. (**A**,**B**). N2a cells were treated with (+) or without (–) Aβ_1-40_ peptide, 10 µM AMF, and the lysosomal inhibitor, leupeptin (Leup) for 3 h. For a subset of the cells (washout), the extracellular TAMRA-Aβ_1-40_ and the agents were removed, and the cells were further incubated for 3 h. (**A**). Cells were harvested, lysed and equal amounts of the protein samples (10 µg/lane) were separated by SDS-PAGE, and subjected to immunoblotting with antibodies specific for Aβ_1-40_ and α-tubulin. (**B**). The intensity of the intact (4 kDa) Aβ_1-42_ signal was quantified and normalized to α-tubulin. (**C**). The Aβ_1-40_ uptake assay as described above was performed with TAMRA-Aβ_1-40_, and the peptide remained in the culture medium was measured. Data indicate mean ± S.E.M. (*n* = 4). *: *p* < 0.05 as assessed by two-way ANOVA followed by Tukey’s multiple comparison tests.

**Figure 5 ijms-23-05885-f005:**
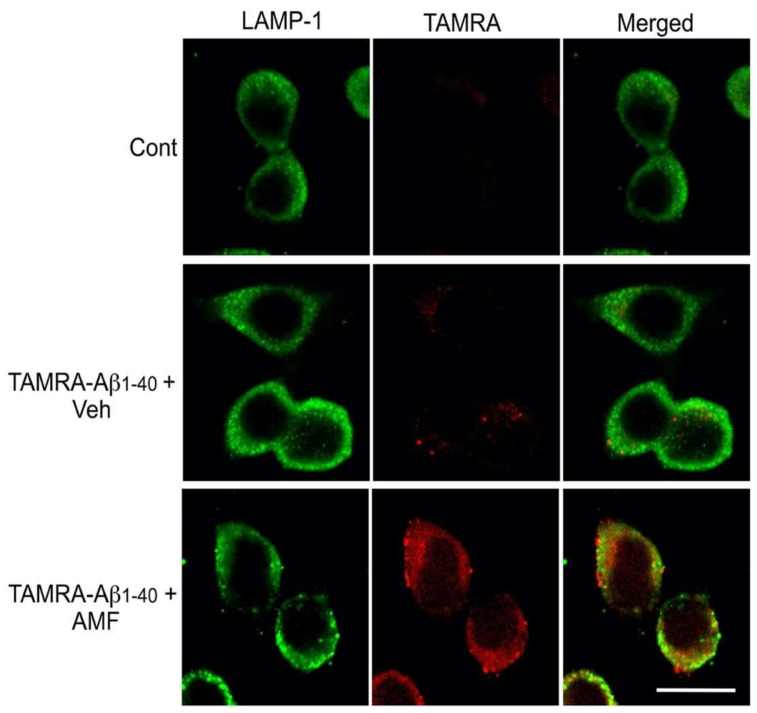
Confocal images reveal localization of the internalized Aβ_1-40_ in lysosomes. N2a cells were incubated with the uptake medium (control) or 200 nM TAMRA-Aβ_40_ in the presence or absence of 10 μM amentoflavone (AMF). Three hours later, cells were fixed with 4% paraformaldehyde and subjected to immunofluorescent labeling with the lysosomal marker LAMP-1 and confocal microscopy. Photographic images of LAMP-1 (green) and TAMRA-Aβ_40_ (red) were captured in the same field and merged. Scale bar: 10 μm.

**Figure 6 ijms-23-05885-f006:**
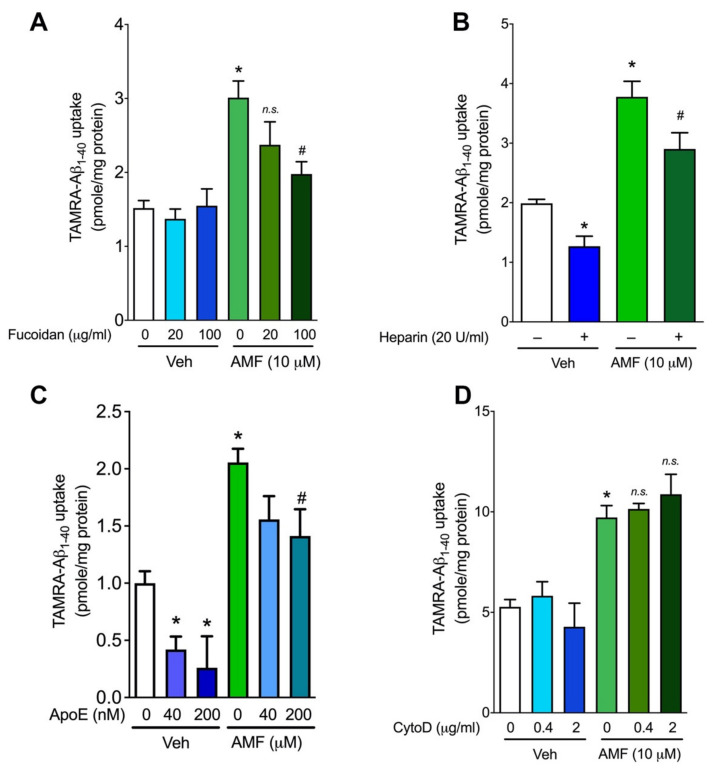
Evidence that amentoflavone increases the receptor-mediated endocytosis of Aβ_1-40_. N2a cells were incubated with or without 10 μM amentoflavone (AMF) in the presence or absence of the scavenger receptor blocker, fucoidan (**A**), the heparan sulfate peptidoglycan inhibitor, heparin (**B**), apolipoprotein E (ApoE) particles (**C**), and the pinocytosis inhibitor, cytochalasin D (**D**). After three hours of incubation, cells were harvested, and the fluorescent intensity of TAMRA-Aβ_1-40_ was determined by fluorometry. Data indicate mean ± S.E.M. from three independent experiments. *: *p* < 0.05 vs. the vehicle group without AMF treatment; #: *p* < 0.05 vs. the amentoflavone-treat group without inhibitors as assessed by one-way ANOVA followed by Tukey’s multiple comparison tests.

**Figure 7 ijms-23-05885-f007:**
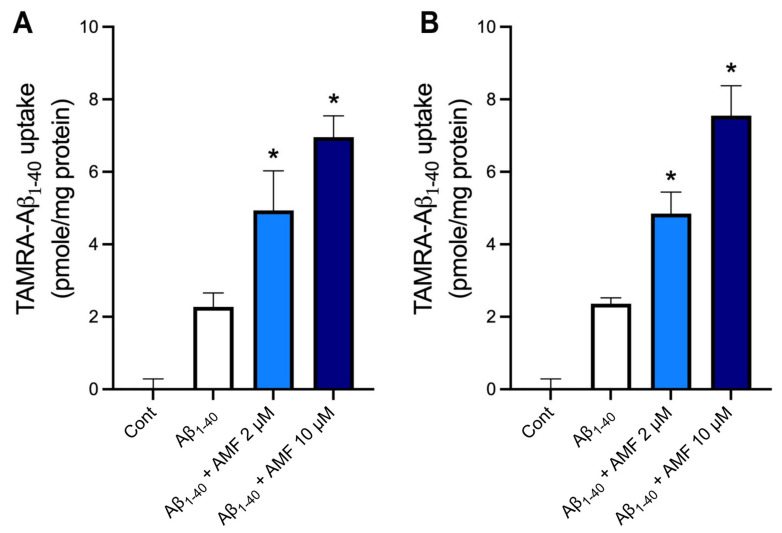
Amentoflavone promotes the cellular uptake of Aβ peptide in glial cells. Mouse BV2 microglial cells (**A**) and immortalized astrocytes (**B**) were incubated with or without 2 or 10 μM amentoflavone (AMF) plus 200 nM TAMRA Aβ_1-40_ for three hours at 37 °C. The cells were harvested, and the fluorescent intensity of TAMRA was determined by fluorometry. Data indicate mean ± S.E.M. from three independent experiments. *: *p* < 0.05 vs. Aβ_1-40_ group by one-way ANOVA followed by Tukey’s multiple comparison test.

## Data Availability

Not applicable.

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
