# Peer review of "Amentoflavone Promotes Cellular Uptake and Degradation of Amyloid-Beta in Neuronal Cells"

_ijms, 2022, doi:10.3390/ijms23115885_

Round 1

Reviewer 1 Report

Han et al. have studied the effects of amentoflavone on the update of amyloid β-protein in the mouse neuronal cells, N2a. The manuscript is written well, and the experiments are executed well. I have few suggestions and hope that these will the authors improve the manuscript.

  1. Please note that the WHO-designated nomenclature for Aβ is “amyloid β-protein.” Please ‎use this ‎in the text to introduce Aβ first and the use the abbreviation. ‘Aβ’ as an abbreviation is fine.‎
  2. Despite being the commonly used and commonly accepted “misnomers,” the eponymous ‎‎term “Alzheimer’s Disease” (AD) is logically incorrect. ‎Historically, the disease was ‎discovered by Alois Alzheimer; the disease was not “his own” disease. Because of the ‎eponymous ‎convention, using the possessive form (apostrophe plus “s” or genitive “s”) is ‎wrong but ‎has been perpetuated in the English scientific literature by our great peers. ‎Many though ‎have avoided it. The Australian Manual of Scientific Style and The Chicago ‎Manual of ‎Style recommend against the use of the possessive form. I suggest taking their ‎editorial ‎advice and applying it throughout the text.
  3. Some minor English issues plague the text. See line 34 for number agreement, “levels … are …” instead of “levels … is …” See line 41 for the missing “of” after “95%.” In line 58, change “discern” to “distinct;” “discern” is a verb, not an adjective. In line 391 and throughout the text, use italicized letter g and the multiplication sign (instead of the letter x) to specify g-force. In line 405, change “spined” to “spun” or “centrifuged”, and “Triplicated” to the noun form “Triplicate.” In 434, change “with” to “using” and remove “a” after “with.” In line 435, change “host-” to prefix “post-“.
  4. In the methods, please specify how concentration of Aβ measured for experiments. Please explain the rationale for the method used to measure the concentration of Aβ. Please also mention how the commercially available Aβ conjugates were characterized or tested in the laboratory that the conjugation by AnaSpec had actually worked. Did the researchers double-test the conjugated forms of Aβ?
  5. In line 94, change “morphologically” to “cytologically”.
  6. Please revise the concentration values of amentoflavone used in Figure 1B. The text in line 91 mentions 1, 3, or 10 μM, but the figure label shows 0, 2, 10, and 50 μM. Which one is correct? Besides, revise the use of “and” instead of the correct use of “or” for logic. For example, in line 91 where the concentrations are listed, “or” should be used instead of “and”. Revise throughout the text for logic.
  7. Some information is missing in the section describing methods. For example, the authors did not mention where they obtained HJ2 from. The composition and concentration of the chemicals in PBS should be described. Describe and add details and rationales on how the membrane-bound Aβ was measured (see lines 161–163). Please supply the relevant details on how the inhibitors were used and where they were sourced from. This includes fucoidan, the ligand for the scavenger receptor class A (MARCO and SRA1), heparin, and ApoE particles.
  8. I may have misunderstood one aspect of the word discussed in the discussion. The authors seem to suggest that amentoflavone promotes degradation of Aβ by the lysosomes. The results do not seem to support such a conclusion; only colocalization with lysosomes was shown. Thus, degradation of Aβ by lysosomes could be independent of presence of amentoflavone or enhancement of Aβ uptake by amentoflavone. This aspect of the discussion should be worded clearly so the reader is not misled.
  9. I am surprised that the authors have used quercetin and EGCG as controls in some of their experiments. Both the chemicals have been recognized as pan-assay interfering compounds (PAINS). I suggest the authors consider the PAINS literature and discuss them. I also suggest that the authors should consult the PAINS-remover database to test amentoflavone, ensuring that amentoflavone of similar molecules used in this study do not have the characteristics of a PAINS molecule. Discuss and present such analyses to strengthen the manuscript. The PAINS database is here: https://www.cbligand.org/PAINS/. See https://doi.org/10.1016/j.jmgm.2017.12.020.

Thank you and good luck for revising your manuscript.

Author Response

The reviewers have raised some excellent points that certainly improved our manuscript. We were encouraged by the positive comments, including: The manuscript is written well, and the experiments are executed well.”, and “The manuscript is well-written and well-designed. The results are interesting and would be of general interest to this field of research”.  We have revised the manuscript and new changes/edits from the original submission have been marked using the “Track Changes” function. We are confident that all the issues raised by the reviewers have been fully addressed. Our detailed responses are as follows:

  1. “Please note that the WHO-designated nomenclature for Aβ is “amyloid β-protein.” Please ‎use this ‎in the text to introduce Aβ first and the use the abbreviation. ‘Aβ’ as an abbreviation is fine”.‎

Response: We thank the reviewer for this comment. We have made the suggested correction in new lines 10 and 31 of the revised manuscript.

  1. Despite being the commonly used and commonly accepted “misnomers,” the eponymous ‎‎term “Alzheimer’s Disease” (AD) is logically incorrect. ‎Historically, the disease was ‎discovered by Alois Alzheimer; the disease was not “his own” disease. Because of the ‎eponymous ‎convention, using the possessive form (apostrophe plus “s” or genitive “s”) is ‎wrong but ‎has been perpetuated in the English scientific literature by our great peers. ‎Many though ‎have avoided it. The Australian Manual of Scientific Style and The Chicago ‎Manual of ‎Style recommend against the use of the possessive form. I suggest taking their ‎editorial ‎advice and applying it throughout the text.

Response: Thank you for the reviewer’s valid point. We have made changes as the reviewer suggested throughout the text of the revised manuscript. However, we would like to follow the editor’s decision on this matter as we see that “Alzheimer’s” is still being predominantly used in recent articles published in IJMS.

  1. “Some minor English issues plague the text. See line 34 for number agreement, “levels … are …” instead of “levels … is …” See line 41 for the missing “of” after “95%.” In line 58, change “discern” to “distinct;” “discern” is a verb, not an adjective. In line 391 and throughout the text, use italicized letter g and the multiplication sign (instead of the letter x) to specify g-force. In line 405, change “spined” to “spun” or “centrifuged”, and “Triplicated” to the noun form “Triplicate.” In 434, change “with” to “using” and remove “a” after “with.” In line 435, change “host-” to prefix “post-“. 

Response: Thank you for finding out those typos and grammatical flaws. All changes have been made as the reviewer suggested.

  1. “In the methods, please specify how concentration of Aβ measured for experiments. Please explain the rationale for the method used to measure the concentration of Aβ. Please also mention how the commercially available Aβ conjugates were characterized or tested in the laboratory that the conjugation by AnaSpec had actually worked. Did the researchers double-test the conjugated forms of Aβ?”

Response: We have provided more details on the preparation of fluorescently conjugated Aβ peptides purchased from AnaSpec, and on how to calculate the Aβ concentrations in the Materials and Methods in new lines 436-441 and 461-463. The vendor provided us the quality control data indicating the purity of the synthesized, conjugated peptide was greater than 95% by HPLC analysis. Typically, we made 500-times concentrated stock solutions of the fluorescent labeled peptide and measured their actual concentration and fluorescent intensity to a plot FU (fluorescent unit) vs peptide concentration curve. Once we measured the fluorescent intensity of samples, we could convert the fluorescent unit to the amount of Aβ. Since our observation that amentoflavone increases the cellular uptake of Aβ was novel, we extensively verified this finding using different types of Aβ species and fluorescent conjugates, and unlabeled Aβ in the present study (Figures 2 and 4). Please note that there was virtually no cellular Aβ uptake when the reverse (scrambled) FAM-Aβ42-1 was used in our assay (Figure 2C), and the conjugated Aβ uptake was markedly blocked by an anti-Aβ antibody. These results clearly indicate that the cellular Aβ uptake is a specific process driven by Aβ peptide but not by any artifact in the assay system.

  1. “In line 94, change “morphologically” to “cytologically”.

Response: This issue has been fixed in new line 107. Thank you.

  1. “Please revise the concentration values of amentoflavone used in Figure 1B. The text in line 91 mentions 1, 3, or 10 μM, but the figure label shows 0, 2, 10, and 50 μM. Which one is correct? Besides, revise the use of “and” instead of the correct use of “or” for logic. For example, in line 91 where the concentrations are listed, “or” should be used instead of “and”. Revise throughout the text for logic”.

Response: All the points raised by the reviewer have been fixed. First, we confirmed that the concentrations in the labels of Figure 1B and C are correct. Second, the concentrations in new line 104 must have been referring to Figure 1C and it has been corrected. Lastly, “and” in the drug concentrations has been replaced with “or” throughout the revised manuscript.

  1. Some information is missing in the section describing methods. For example, the authors did not mention where they obtained HJ2 from. The composition and concentration of the chemicals in PBS should be described. Describe and add details and rationales on how the membrane-bound Aβ was measured (see lines 161–163). Please supply the relevant details on how the inhibitors were used and where they were sourced from. This includes fucoidan, the ligand for the scavenger receptor class A (MARCO and SRA1), heparin, and ApoE particles.

Response: We agree with the reviewer’s suggestion and all issues have been resolved. First, the source of HJ2 and ApoE particles (originally stated in the Acknowledgement section) has been described in the Materials and Methods in new lines 412-413. Second, the composition of PBS has been fully provided in new lines 450-451. Third, we have now provided rational and more details for the measurement of membrane-bound form of Aβ in line 451. The origin of inhibitors and details on how we treated cells with them have now been provided in new lines 414-415 and 464-471, respectively.

  1. I may have misunderstood one aspect of the word discussed in the discussion. The authors seem to suggest that amentoflavone promotes degradation of Aβ by the lysosomes. The results do not seem to support such a conclusion; only colocalization with lysosomes was shown. Thus, degradation of Aβ by lysosomes could be independent of presence of amentoflavone or enhancement of Aβ uptake by amentoflavone. This aspect of the discussion should be worded clearly so the reader is not misled.

Response:  We apology for confusion. We have revised the problematic phrase to clarify our findings and interpretations as follows: In the present study, we report that amentoflavone promotes cellular uptake of Aβ probably via the class A scavenger receptors. The internalized Aβ, in turn, undergoes lysosomal degradation without causing cytotoxicity in neuronal cells” in 393-396.

  1. I am surprised that the authors have used quercetin and EGCG as controls in some of their experiments. Both the chemicals have been recognized as pan-assay interfering compounds (PAINS). I suggest the authors consider the PAINS literature and discuss them. I also suggest that the authors should consult the PAINS-remover database to test amentoflavone, ensuring that amentoflavone of similar molecules used in this study do not have the characteristics of a PAINS molecule. Discuss and present such analyses to strengthen the manuscript. The PAINS database is here: https://www.cbligand.org/PAINS/. See https://doi.org/10.1016/j.jmgm.2017.12.020. Thank you and good luck for revising your manuscript.

Response: We appreciate the reviewer for bringing up this point. First of all, we have verified that our findings in the present study is very unlikely due to false positive reactions of amentoflavone because it does not contain “a PAINS motif”. In our screening assay presented in Figure 1, we chose as negative controls quercetin and EGCG since they are relatively well studied polyphenols demonstrating their anti-amyloid effects. Even though they contain a PAINS motif which is catechol, they did not interfere with the Aβ uptake assay in our experimental system (Figure 1C). However, amentoflavone and its similar biflavonoids, which do not contain any potential PAINS motif based on the database search provided by the reviewer, did work in our assay system to increase the Aβ uptake assay. This issue has been discussed in new lines 314-324, which would strengthen our manuscript. Thank you again for the important input.

Reviewer 2 Report

In this manuscript, Han and colleagues report that amentoflavone promotes cellular uptake and degradation of amyloid-beta in neuronal cells. The manuscript is well-written and well-designed. The results are interesting and would be of general interest to this field of research. However, some points need to be addressed before publication.

Major points

  • The Authors wrote they analyzed data by using one-way or two- way ANOVA followed by a host-hoc multiple comparison (which one?). This is the correct approach (considering a normal distribution of the data) that, however, has not been properly applied by the Authors. In particular, in figure 1, the Authors wrote they used one way ANOVA. This is not correct because the factors are two: concentration x treatment (compounds). In figure 3, the Authors wrote they used a t-test. This is unexplainable. Indeed, the factors are treatment x time, and thus the correct way is a repeated measure two- way ANOVA. The Authors need to re-analyze the data by using the correct statistical design that depends on the factors need to be analyzed.

Minor points

  • The hypothesis tested by the Authors and also the rationale of this study is partially reported in the introduction (line 68-71) but not in the abstract.

  • As reported above, the Authors should add which post-hoc they used.

  • Because amentoflavone belongs to the family of polyphenols, it is important to discuss and report the potential therapeutic benefits that can be obtained by using polyphenols for the treatment of neurological disorders such as AD. In this respect, the Authors could add and discuss the following recent review (PMID: 34624428).

Author Response

The reviewers have raised some excellent points that certainly improved our manuscript. We were encouraged by the positive comments, including: The manuscript is written well, and the experiments are executed well.”, and “The manuscript is well-written and well-designed. The results are interesting and would be of general interest to this field of research”.  We have revised the manuscript and new changes/edits from the original submission have been marked using the “Track Changes” function. We are confident that all the issues raised by the reviewers have been fully addressed. Our detailed responses are as follows:

Major points

  1. “The Authors wrote they analyzed data by using one-way or two- way ANOVA followed by a host-hoc multiple comparison (which one?). This is the correct approach (considering a normal distribution of the data) that, however, has not been properly applied by the Authors. In particular, in figure 1, the Authors wrote they used one way ANOVA. This is not correct because the factors are two: concentration x treatment (compounds). In figure 3, the Authors wrote they used a t-test. This is unexplainable. Indeed, the factors are treatment x time, and thus the correct way is a repeated measure two- way ANOVA. The Authors need to re-analyze the data by using the correct statistical design that depends on the factors need to be analyzed”.

Response: Thank you for the reviewer’s valid point. We have reassessed our static analysis and made corrections as follows. First, we have now specified the post-hoc multiple comparison as “Tukey’s” throughout the revised manuscript in new lines 125, 154, 221, 262, 276, and 511. Second, we initially performed a one-way ANOVA test for the data presented in Figures 1A and 1B since we compared the treatment groups to their own control. However, we agree with the reviewer that two-way ANOVA should be used for these data based on the fact that there are two independent variables. We have corrected this issue in new Figures 1B and C.

Minor points

  1. “The hypothesis tested by the Authors and also the rationale of this study is partially reported in the introduction (line 68-71) but not in the abstract”.

Response: Thank you for the reviewer’s comment. We have now stated the rationale of the study in the Abstract in new lines 14-16.

  1. “As reported above, the Authors should add which post-hoc they used”.

Response: Please see our response above.

  1. Because amentoflavone belongs to the family of polyphenols, it is important to discuss and report the potential therapeutic benefits that can be obtained by using polyphenols for the treatment of neurological disorders such as AD. In this respect, the Authors could add and discuss the following recent review (PMID: 34624428)”.

Response: We appreciate the reviewer for bringing this to our attention. Many studies in animal models and humans suggest the beneficial effects of natural polyphenols for AD therapy probably through their antiinflammatory and antioxidant activities. We and others have demonstrated that amentoflavone is an anti-inflammatory and anti-oxidative biflavonoid that can attenuate neuronal cell death and cognitive decline in cell culture and rodent models of AD (please the references cited in the manuscript). However, we found the amentoflavone’s action on promoting cellular uptake of Ab in the present study is very unlikely due to their antiinflammatory and antioxidative effects. We have shown that potent antioxidant polyphenols such as quercetin and EGCG have no effects on the cellular Ab uptake in our experimental system (Figure 1C), indicating the unique action of amentoflavone. That was a major reason why we did not attempt to generalize our findings to other types of polyphenolic compounds at this time. We believe amentoflavone (but not other similar biflavonoids, monoflavonoid, or other polyphenols) uniquely binds to Ab peptide and causes conformational change in Ab structure, resulting an increase in the cellular transport Ab via certain cell-surface scavenger receptors. We are currently working on the follow-up studies to further explore this hypothesis and the effects of other classes of polyphenols on amyloid clearance in vitro and in vivo. We have briefly discussed these issues with providing the refence (PMID: 34624428) as the reviewer suggested along with other recent reviews on polyphenols and neurological disorders in new lines 399-406.

Round 2

Reviewer 2 Report

The Authors have successfully addressed all my comments. Well done!